# LOGOS: PRECISION RETRIEVAL VIA LOGICAL DOCUMENT GRAPHS FOR RETRIEVAL-AUGMENTED GENERATION

## ABSTRACT

Retrieval-Augmented Generation (RAG) systems struggle with long documents because conventional retrieval methods provide noisy, page-level context that degrades generation quality. These methods are fundamentally limited by treating documents as a linear sequence of pages, which breaks the crucial logical dependencies (e.g., tables, paragraphs, and references) that span across page boundaries. To overcome this limitation, we propose Precision Retrieval via Logical Document Graphs for Retrieval-Augmented Generation (LOGOS), a new RAG method that achieves precision retrieval by modeling a document's intrinsic logical structure. LOGOS transforms a document into a graph where semantic regions are nodes and logical connections are edges, effectively bridging page breaks. A Graph Neural Network then generates fine-grained, context-aware representations for each node, enabling a more concise and semantically relevant context for the generator. Experiments on the ViDoRe and MMDOCIR benchmarks show that LOGOS sets a new state-of-the-art, significantly outperforming strong baselines by up to 2% in average Recall@1. For the reproducibility of results, we have made the code of LOGOS available anonymously at: `https://anonymous.4open.science/r/LOGOS-19048-637D`.

## 1 INTRODUCTION

With the acceleration of digital transformation, a vast corpus of documents—including academic papers, technical manuals, financial reports, and legal contracts—has become the primary medium for knowledge dissemination and information exchange. The challenge of efficiently and accurately tapping into this knowledge has given rise to Retrieval-Augmented Generation (RAG), a paradigm that enhances large language models by grounding them in external, up-to-date information (Lewis et al., 2020). RAG has quickly become a critical research area, as it directly addresses key limitations in standalone models, such as factual inaccuracies and outdated knowledge, thereby increasing the reliability and applicability of generation systems (Gao et al., 2023; Huang & Huang, 2024). The recent advent of Multimodal Large Language Models (MLLMs) has brought revolutionary advancements, showcasing a powerful ability to process and reason over diverse modalities like text, images, and tables simultaneously.

As illustrated in the performance benchmarks of Qwen2.5-VL-7B (Fig.1.a), this challenge is clearly quantified. The initial performance climb from a context window of 256 to 8k underscores the absolute necessity of providing sufficient context beyond a single page. However, the subsequent, sharp performance decay from 8k up to 128k reveals the critical limitation of simply extending context length. This trend demonstrates a clear "cost of noise": retrieving an excessive number of passages, a common outcome of page-level chunking, dilutes the context and ultimately leads to a significant degradation in downstream task accuracy while linearly increasing computational costs (Shi et al., 2023; Amiraz et al., 2025). The limited ability of models to effectively handle ever-increasing context lengths is the primary motivation for our method, which aims to precisely retrieve and process relevant information scattered throughout the document, rather than relying on brute-force context expansion.

In practice, the logical flow of information frequently traverses the pages. For instance, a chapter title may appear at the bottom of one page with its corresponding body text beginning on the next; a textual

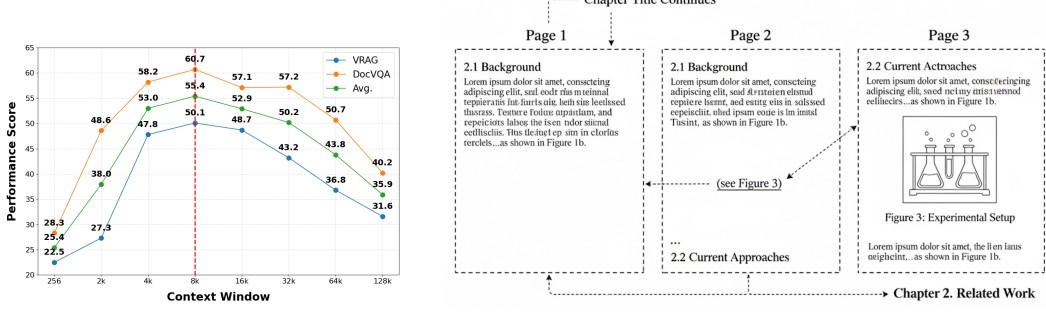

(a) Performance of Document-based QA        (b) Illustration of Broken Logical Dependency

Figure 1: The challenge in conventional RAGs. (a) A quantitative analysis in long-document QA. Performance peaks with a medium context and then declines. (b) An example of a logical dependency being broken by naive page-level processing.

reference may point to a figure located several pages away (illustrated in Fig.1.b); a single large table or a long list can naturally span across multiple consecutive pages; and a complex argument or a single coherent paragraph is often split between the end of one page and the beginning of another. The page-level processing paradigm fails to capture these cross-page dependencies, leading to a significant loss of contextual information. This, in turn, limits the model's ability to form a global semantic understanding of the document, thereby degrading the performance of downstream tasks like question answering, information extraction, and summarization. Although some efforts attempt to mitigate this by using larger context windows or simple text concatenation (Beltagy et al., 2020), these methods struggle to effectively model the complex, nonlinear structure of documents (Liu et al., 2024). Recent studies show that even with vastly expanded context windows, models still face challenges with long-range dependencies and information retrieval accuracy, especially as the input length increases (Li et al., 2025; Wang et al., 2024). More recent studies, such as Colpali (Faysse et al., 2025) and DSE (Ma et al., 2024), have begun to explore efficient retrieval directly from document images using vision-language models, signaling a promising direction for overcoming these challenges. Our experiments include these methods as key baselines. While we do not directly compare against older architectures like Longformer, we instead benchmark against stronger, more recent text-based models such as DPR-Phi3 (Dong et al., 2025) and Col-Phi3 (Dong et al., 2025), ensuring a robust and fair evaluation against the current state-of-the-art.

To address these challenges, we introduce a novel framework named Logical Document Graphs for Retrieval-Augmented Generation (LOGOS). LOGOS is to transform the document's physical page-based structure into a graph structure that better reflects its intrinsic logical organization. Instead of treating pages as the fundamental unit, the proposed method LOGOS delves deeper into the semantic regions within each page (e.g., paragraphs, images, tables). LOGOS achieves this by constructing a Cross-Page Heterogeneous Graph, which explicitly connects all semantically related regions across the entire document, regardless of their page location. This graph structure simultaneously captures intra-page spatial layouts and inter-page logical relationships. Building on this foundation, this architecture of proposed LOGOS leverages the powerful capabilities of Graph Neural Networks (GNNs) to propagate and aggregate information across the entire document graph. The leverage of GNNs the representation of each semantic region to be enriched by its multi-hop neighbors, resulting in a global, context-aware representation. The main contributions of this work are threefold:

- We propose a novel method LOGOS for constructing a cross-page heterogeneous graph that transcends page boundaries to explicitly model long-range dependencies and logical connections within a document.

- We design a GNN-based information fusion mechanism that effectively integrates the structural, semantic, and multimodal information of a document to generate high-quality, context-aware representations. We conduct experiments on two widely used benchmarks, ViDoRe and MMDOCIR, to validate the effectiveness of our approach. LOGOS sets a new state-of-the-art by outperforming a wide range of strong baselines, including both dense

retrieval and late-interaction models, achieving significant improvements of up to 2% in average Recall@1.

## 2 RELATED WORK

Methodologies for document retrieval have evolved from classical sparse methods like BM25 (Robertson & Zaragoza, 2009) to modern dense retrieval architectures. Models like Dense Passage Retriever (DPR) (Karpukhin et al., 2020) marked a significant advancement by using deep neural networks to perform semantic search, but their reliance on breaking documents into arbitrary text "chunks" often disrupts the natural logical flow. To address the limitations of text-only retrieval on visually rich documents, new benchmarks and methods were developed in parallel. The ViDoRe benchmark, for instance, was one of the first to establish a large-scale evaluation for document-level understanding, pushing models beyond isolated passages to comprehend entire pages. In response, a new wave of vision-language models has emerged. Approaches like DSE (Ma et al., 2024) and ColPali (Faysse et al., 2025) leverage the power of multimodal encoders to process entire document pages directly from their rendered images, preserving crucial visual and layout cues. These methods, inspired by late-interaction architectures like ColBERT (Khattab & Zaharia, 2020), were often evaluated on newer benchmarks like MMDocIR (Dong et al., 2025). MMDocIR further advanced the field by introducing a fine-grained evaluation that acknowledged distinct layout regions within pages. However, a key gap remained: while the benchmark provided the framework for region-level analysis, the retrieval methods themselves still predominantly operated at the page level. They lacked a targeted mechanism to either exploit these intra-page structures or, more critically, to connect them across different pages. Our work, LOGOS, directly addresses this gap by constructing a global, cross-page graph to capture a document's true logical structure, enabling a more profound and contextually aware understanding.

## 3 OUR APPROACH

We propose Logical Document Graphs for Retrieval-Augmented Generation (LOGOS), a framework designed to solve the critical problem of context fragmentation in RAG systems that process long documents. Conventional retrieval methods, including recent page-level VLM approaches like ColPali and DSE, are fundamentally limited by treating documents as a linear sequence of pages. This paradigm operates at a coarse granularity and inevitably breaks the crucial logical dependencies—such as paragraphs, tables, or references—that span across page boundaries, leading to a loss of context and lower retrieval precision. LOGOS overcomes this limitation by transforming a document's physical page structure into a global, cross-page graph that represents its intrinsic logical organization. The overall pipeline is illustrated in Figure 2 and comprises three core stages: (a) Document Split, (b) Cross-Page Heterogeneous Graph Construction, and (c) GNN Augmented Retrieval.

### 3.1 DOCUMENT SPLIT

The initial step in current RAG pipeline is to chunk the document. Instead of employing naive strategies like fixed-size or page-level chunking, which arbitrarily sever logical units, we adopt a sophisticated layout-aware approach. Our design decision is to leverage a state-of-the-art, end-to-end model to preserve the semantic integrity of the document's components with high accuracy. To this end, we employ the Vision Grid Transformer (VGT) (Lee et al., 2023), a non-autoregressive transformer-based model for document layout analysis. Unlike traditional methods that may process documents sequentially or rely on separate stages for detection and classification, VGT analyzes the entire page image holistically. It divides the page into a grid, processes grid cell embeddings with a transformer architecture, and simultaneously predicts the bounding boxes and class labels of all layout components in a single pass. This allows the model to accurately identify and classify semantic regions by capturing global layout context.

Formally, the output of this VGT-driven stage is a set of semantic regions $S = \{s_1, s_2, \ldots, s_N\}$, where $N$ is the total number of regions in the document. Each region $s_i \in S$ is a tuple $(c_i, \tau_i, b_i)$, containing its raw content $c_i$ (text or image data extracted from the region), its semantic type $\tau_i \in \{\text{paragraph}, \text{table}, \text{image}, \ldots\}$, and its bounding box coordinates $b_i = (x_{i1}, y_{i1}, x_{i2}, y_{i2})$. This fine-grained and structurally accurate segmentation forms the foundation for our graph.

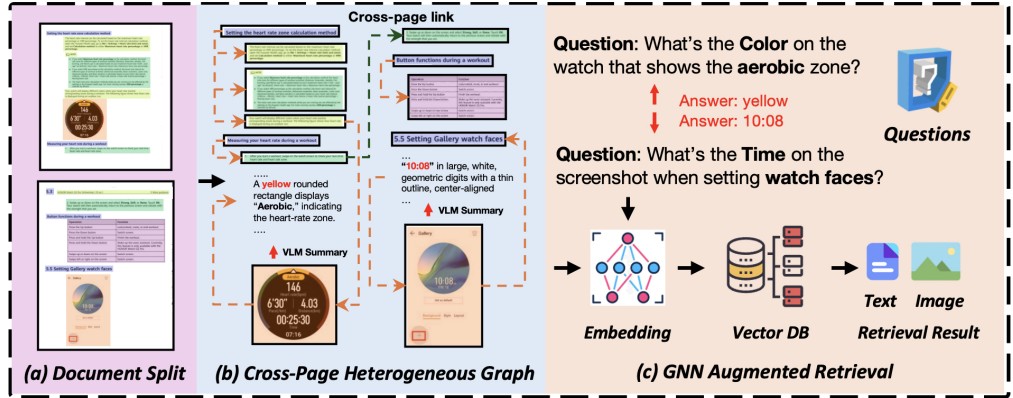

Figure 2: **The LOGOS Pipeline.** The process begins with **(a) Document Split**, where a layout-aware model segments each page into semantic regions. Next, in the **(b) Cross-Page Heterogeneous Graph Construction** stage, these nodes are connected across pages based on logical relationships. Finally, during **(c) GNN Augmented Retrieval**, a GNN processes the entire graph to generate context-aware embeddings for each node.

## 3.2 CROSS-PAGE HETEROGENEOUS GRAPH CONSTRUCTION

This stage transforms the flat set of semantic regions $S$ into a rich, structured graph that captures both local and long-range dependencies.

**Graph Definition.** We formally define a document as a heterogeneous graph $G = (V, E)$, where $V$ is the set of nodes corresponding to the semantic regions $S$, and $E$ is the set of typed edges. Let $\mathcal{R} = \{\text{adj}, \text{sim}, \text{cont}, \text{ref}\}$ be the set of relation types.

**Multimodal Node Feature Enhancement.** To create a unified representation space, we must encode both textual and visual nodes. For each visual node $v_i$ (where its type $\tau_i \in \{\text{image}, \text{table}\}$), we employ Qwen2.5-VL to generate a structured textual description $d_i = f_{\text{VLM}}(c_i)$. This description captures key visual attributes. This design choice allows us to unify all nodes into a common textual modality. A shared text encoder, $f_{\text{enc}}$, then computes the initial feature vector $\mathbf{h}_i^{(0)} \in \mathbb{R}^d$ for each node $v_i$:

$$\mathbf{h}_i^{(0)} = \begin{cases} f_{\text{enc}}(c_i) & \text{if } \tau_i = \text{paragraph} \\ f_{\text{enc}}(d_i) & \text{if } \tau_i \in \{\text{image}, \text{table}\} \end{cases} \tag{1}$$

where $d$ is the embedding dimension. This equation explicitly assigns the encoded representation to the initial hidden state $\mathbf{h}_i^{(0)}$ for each node.

**Cross-Page Edge Formulation.** The cornerstone of LOGOS is its ability to model relationships beyond page boundaries. We formally define the neighbor set $\mathcal{N}_i^r$ for each node $v_i$ and relation type $r \in \mathcal{R}$ as follows:

- **Spatial Adjacency** ($r = $ **adj**): Edges connect spatially consecutive regions on the same page. To determine the reading order, nodes on a given page are sorted first by their top vertical coordinate ($y_1$) and then by their left horizontal coordinate ($x_1$). This top-to-bottom, left-to-right heuristic establishes a directed edge from each node to the one immediately following it. The neighbor set is thus defined as $\mathcal{N}_i^{\text{adj}} = \{v_j \mid v_j$ is the node immediately succeeding $v_i$ in the reading order on the same page$\}$.
- **Semantic Similarity** ($r = $ **sim**): To connect related but non-adjacent concepts, we create undirected edges. The neighbor set is formally defined as $\mathcal{N}_i^{\text{sim}} = \{v_j \mid \text{cos\_sim}(\mathbf{h}_i^{(0)}, \mathbf{h}_j^{(0)})$ is among the top-$k$ scores for node $v_i, \forall v_j \in V, j \neq i\}$.
- **Cross-Page Continuation** ($r = $ **cont**): An edge explicitly connects the last semantic region of page $P_k$ to the first semantic region of page $P_{k+1}$. While this simple heuristic may not

perfectly handle cases where a paragraph splits mid-sentence, it provides a robust structural prior for maintaining logical flow across page breaks. The neighbor set is $\mathcal{N}_i^{\text{cont}} = \{v_j \mid v_i$ is the last node in reading order on page $P_k$ and $v_j$ is the first node on page $P_{k+1}\}$.

- **Explicit Reference** ($r = \textbf{ref}$): To capture direct structural links, we parse the text content of nodes to find explicit textual cues (e.g., "see Figure 3", "in Table 2"). We use regular expressions to identify these references and create directed edges from the text node to the corresponding figure or table node. The neighbor set is defined as $\mathcal{N}_i^{\text{ref}} = \{v_j \mid$ the content $c_i$ contains a textual reference to node $v_j\}$.

By defining these edges, particularly the cross-page links, the graph transcends the physical page structure to reflect the document's true logical topology.

### 3.3 GNN AUGMENTED RETRIEVAL

**Contextual Information Fusion.** With the graph constructed, we employ a Graph Neural Network (GNN) to generate context-aware embeddings. A GNN is the natural choice for this task, as it enables information to propagate along the defined edges, allowing each node's representation to be enriched by its multi-hop neighborhood. Given the heterogeneous nature of our graph, we use a Relational Graph Convolutional Network (R-GCN). The embedding $\mathbf{h}_i^{(l+1)}$ for a node $v_i$ at layer $l+1$ is updated by aggregating messages from its neighbors under different relation types:

$$\mathbf{h}_i^{(l+1)} = \sigma \left( \mathbf{W}_0^{(l)} \mathbf{h}_i^{(l)} + \sum_{r \in \mathcal{R}} \sum_{j \in \mathcal{N}_i^r} \frac{1}{|\mathcal{N}_i^r|} \mathbf{W}_r^{(l)} \mathbf{h}_j^{(l)} \right) \tag{2}$$

where $\mathcal{R}$ is the set of defined relation types, $\mathcal{N}_i^r$ is the formally defined set of neighbors of node $v_i$ under relation $r$, $\mathbf{W}_r^{(l)} \in \mathbb{R}^{d \times d}$ is a relation-specific learnable weight matrix, $\mathbf{W}_0^{(l)}$ is the self-loop weight matrix, and $\sigma$ is a non-linear activation function. By stacking $L$ such layers, each node's final embedding $\mathbf{h}_i^{(L)}$ captures structural and semantic information from its $L$-hop neighborhood, effectively integrating global document context.

**Node-Level Retrieval with Enhanced Embeddings.** The final, context-aware node embeddings, $\mathbf{h}_i^{(L)}$, serve as the basis for our precise, fine-grained retrieval mechanism. This process is divided into offline indexing and online querying.

- **Offline Indexing**: For each document in the corpus, the final, GNN-enhanced embedding $\mathbf{h}_i^{(L)}$ for every node $v_i$ (representing paragraphs, images, tables, etc.) is computed and indexed in a vector database. Each vector is stored alongside a reference to its original node content $c_i$.

- **Online Querying**: Given a user query $q$, it is first encoded using the same text encoder to produce a query vector $\mathbf{q} = f_{\text{enc}}(q) \in \mathbb{R}^d$. A similarity search (e.g., cosine similarity) is performed between $\mathbf{q}$ and all indexed node embeddings in the vector database. The top-$K$ nodes $\{v_{j_1}, \ldots, v_{j_K}\}$ corresponding to the highest similarity scores are retrieved. The original content $\{c_{j_1}, \ldots, c_{j_K}\}$ of these retrieved nodes—which can be a mix of text passages and VLM-generated descriptions for images—is then concatenated and provided as the precise, synthesized context to the LLM generator.

This GNN-augmented, node-level retrieval process ensures that the context provided to the LLM is not only highly relevant but also holistically informed by the document's global logical structure.

## 4 EXPERIMENTS AND RESULTS

In this section, we provide experimental results to demonstrate the effectiveness of the LOGOS framework. We evaluate its performance on two widely used document retrieval benchmarks and compare it to existing methods.

## 4.1 Datasets

**ViDoRe.** Our primary evaluation is performed on the ViDoRe (Faysse et al., 2025) benchmark, which is specifically designed to assess document retrieval systems in matching queries to relevant documents at the page level. ViDoRe's strength lies in its diversity, encompassing a wide range of subtasks that span multiple modalities (text, figures, tables) and thematic domains (e.g., medical, scientific, and administrative). We follow the official protocol, training our model on the provided training split of 118,695 query-page pairs and evaluating on the test set, which contains documents entirely unseen during training to prevent data contamination.

**MMDocIR.** To further assess the fine-grained retrieval capabilities of our model, we also use the MMDocIR benchmark (Dong et al., 2025). MMDocIR focuses on multi-modal retrieval in long, structurally complex documents, featuring ten distinct categories such as academic papers, financial reports, and laws. While a key contribution of this benchmark is its support for region-level evaluation, we adopt the standard page-level retrieval task for our main experiments to ensure a fair and direct comparison against existing state-of-the-art methods that primarily report page-level results.

## 4.2 Metrics.

To measure the performance of our retrieval system, we adopt the standard information retrieval metric of Recall@K. This metric calculates the percentage of queries for which the ground-truth relevant page is successfully found within the top-K retrieved results.

Consistent with prior work (Dong et al., 2025) and the official benchmark protocols, we report results for K=1, K=3, and K=5 to provide a comprehensive assessment of retrieval precision at different cutoffs.

## 4.3 Implementation Details.

We apply Low-Rank Adapters to the transformer layers of the language model, with a scaling factor $\alpha = 32$ and rank $r = 32$, along with a final randomly initialized projection layer. Optimization is performed using the 8-bit paged AdamW optimizer. Training is conducted on an 8-GPU setup with data parallelism. We use a learning rate of $5 \times 10^{-5}$ with linear decay and 2.5% warmup steps, and a batch size appropriate for the available GPU memory and computational budget.

## 4.4 Baselines

We compared LOGOS against several competitive baselines that represent different paradigms in document retrieval. These include methods based on vision-language models, late-interaction architectures, and powerful large language models.

**DSE** (Ma et al., 2024) introduces a novel retrieval paradigm that directly encodes document screenshots into dense representations using a large vision-language model. This approach is distinct because it bypasses traditional text extraction and parsing, thereby preserving all visual and layout information present in the document page, such as text style, images, and spatial structure.

**ColPali and ColQwen** (Faysse et al., 2025) are methods based on a late-interaction retrieval architecture, likely inspired by ColBERT. In this framework, document pages and queries are encoded into multi-vector representations. Relevance is computed through a fine-grained, token-level interaction between the query and document embeddings. These models leverage powerful vision-language models as their backbone encoders, specifically using variants of Pali and Qwen to process multimodal document content.

**DPR-Phi3** (Dong et al., 2025) is a strong text-based retrieval model that adapts the classic Dense Passage Retriever (DPR) dual-encoder architecture. It uses the powerful and efficient Phi-3 language model as its text encoder to generate single-vector representations for both queries and document text. This baseline represents the state-of-the-art in standard dense retrieval using modern small language models.

**Col-Phi3**  (Dong et al., 2025) is another text-based baseline that combines the ColBERT-style late-interaction mechanism with the Phi-3 language model. Unlike DPR-Phi3, it computes relevance scores based on token-level similarities, which often leads to more accurate and robust retrieval results at the cost of higher computational complexity. It serves as a powerful baseline for text-based, fine-grained retrieval.

## 5 EXPERIMENTS AND RESULTS

To validate the effectiveness of our proposed method, we conducted comprehensive experiments on two large-scale benchmark datasets: ViDoRe and MMDOCIR. The experimental results demonstrate that our **LOGOS** model achieves state-of-the-art performance, outperforming existing baseline models across various metrics.

### 5.1 EVALUATION ON VIDORE

We first evaluated our model on the ViDoRe dataset, comparing it against a range of strong baseline methods, including BiPali , DSE (Ma et al., 2024) , ColPali (Faysse et al., 2025), ColQwen (Faysse et al., 2025), Col-Phi3 (Dong et al., 2025) and DPR-Phi3 (Dong et al., 2025). As shown in Table 1, our proposed LOGOS achieves the highest performance on all individual sub-tasks and sets a new state-of-the-art on the average score.

| Method | Source | DocQ | TabF | TATQ | Shift | AI | Gov. | Health. | Avg. |
|--------|--------|------|------|------|-------|-----|------|---------|------|
| BiPali | CVPR 24 | 20.0 | 63.2 | 20.4 | 34.0 | 59.0 | 57.0 | 56.0 | 44.6 |
| DSE | EMNLP 25 | 53.1 | 82.3 | 64.5 | 71.8 | **95.0** | 90.6 | 92.9 | 71.2 |
| ColPali | ICLR 25 | 45.6 | 75.4 | 53.1 | 55.0 | 93.0 | 85.0 | 88.0 | 72.3 |
| ColQwen | ICLR 25 | 45.9 | 76.4 | 58.4 | 57.0 | 93.0 | 86.0 | 93.0 | 74.9 |
| **LOGOS (ours)** | This Paper | **48.3** | **77.5** | **62.6** | **61.0** | 94.0 | **88.0** | **94.0** | **76.9** |

Table 1: Comprehensive evaluation of baseline models and our proposed method on *ViDoRe*. Results are presented using Recall@1 metrics.

Specifically, LOGOS obtains an average Recall@1 score of **76.9%**, which is a improvement over the ColQwen (74.9%) and DSE (71.2%). This highlights the superior performance and generalization capability of our model in handling diverse document-based question-answering tasks. Our model's advantages are consistent across the various sub-domains. For instance, on the TabF and TATQ datasets, which involve complex table-based reasoning, LOGOS achieves scores of **77.5%** and **62.6%**, respectively. Similarly, on domain-specific collections such as Gov. and Health., our model leads with scores of **88.0%** and **94.0%**. These results underscore the effectiveness of our heterogeneous graph-based approach in effectively capturing and leveraging the complex structural and semantic information within documents.

### 5.2 EVALUATION ON MMDOCIR

To further assess the fine-grained retrieval capabilities of our model, we conducted page-level retrieval experiments on the MMDOCIR dataset. As shown in Table 2, our model demonstrates a clear advantage at **Recall@1**. The strongest baseline, Col-Phi3, achieves an average macro score of 57.0%. Our LOGOS model surpasses this, obtaining a new state-of-the-art average macro score of 58.8%. The performance gains are particularly pronounced in several document categories, such as 'Academic Paper' and 'News', where our model achieves scores of 63.8% and 58.2%, respectively. This highlights our model's superior ability to understand complex layouts and unstructured text. While our method shows a slight performance dip on highly regularized documents like 'Government' and 'Laws', its substantial gains in other diverse categories demonstrate its overall robustness.

When the retrieval metric is relaxed to **Recall@5**, all models show improved performance. The best-performing baseline, Col-Phi3, reaches an average macro score of 82.2%. Once again, our LOGOS model sets a new benchmark by achieving a superior score of 84.7%. This indicates that our model is not only precise in identifying the single most relevant page but also highly effective at ranking relevant pages within the top results. The detailed per-category scores, as presented in

| Method | Resear. Report | Admin &Indu. | Tutori.& Worksh. | Acade. Paper | Brochure | Finance Report | Guide-book | Government | Laws | News | Average Macro |
|---|---|---|---|---|---|---|---|---|---|---|---|
| **Recall@k = 1** | | | | | | | | | | | |
| DSE (2024) | 53.0 | 50.0 | 54.0 | 48.7 | 45.1 | 43.0 | 51.5 | 46.9 | 54.2 | 33.6 | 48.0 |
| ColPali (2025) | 56.0 | 51.8 | 58.6 | 55.9 | 52.0 | 47.2 | 57.9 | 53.9 | **64.0** | 32.8 | 53.0 |
| ColQwen (2025) | 56.5 | 51.2 | 57.5 | 58.1 | 53.5 | 48.9 | 59.3 | 56.8 | 63.8 | 44.2 | 55.0 |
| DPR-Phi3 (2025) | 58.9 | 50.4 | 57.4 | 59.0 | 57.3 | 44.6 | 63.8 | 50.5 | 64.4 | 35.0 | 54.1 |
| Col-Phi3 (2025) | 56.7 | 50.4 | 56.9 | 61.3 | 54.8 | 50.7 | 60.8 | **61.3** | 63.6 | 54.0 | 57.0 |
| **LOGOS (Our)** | **59.5** | **52.5** | **59.1** | **63.8** | **58.0** | 51.8 | **64.0** | 60.5 | 63.1 | **58.2** | **58.8** |
| **Recall@k = 3** | | | | | | | | | | | |
| DSE (2024) | 75.4 | 65.0 | 73.9 | 79.8 | 69.5 | 63.5 | 75.4 | 71.5 | 81.4 | 50.4 | 70.6 |
| ColPali (2025) | 77.6 | 71.8 | 79.4 | 83.4 | 72.6 | 66.1 | 80.0 | **80.4** | **86.4** | 49.6 | 74.7 |
| ColQwen (2025) | 78.5 | 73.0 | 78.1 | 84.0 | 71.2 | 66.9 | 79.4 | 79.9 | 84.5 | 59.8 | 75.5 |
| DPR-Phi3 (2025) | 80.3 | 66.5 | 77.6 | 83.9 | 71.9 | 63.8 | 79.8 | 71.4 | 84.5 | 55.5 | 73.5 |
| Col-Phi3 (2025) | 80.2 | 74.1 | 77.4 | 84.8 | 69.1 | 67.7 | 78.7 | 79.5 | 81.8 | 69.3 | 76.3 |
| **LOGOS (Our)** | **82.0** | **76.0** | **80.5** | **86.5** | **75.0** | **69.5** | **82.5** | 78.5 | 84.5 | **74.0** | **78.9** |
| **Recall@k = 5** | | | | | | | | | | | |
| DSE (2024) | 84.0 | 80.2 | 78.7 | 87.0 | 75.7 | 73.0 | 82.0 | 77.3 | 88.3 | 58.4 | 78.5 |
| ColPali (2025) | 84.6 | 79.3 | 82.3 | 89.0 | 79.8 | 72.1 | 86.7 | 84.9 | **92.4** | 56.9 | 80.8 |
| ColQwen (2025) | 85.2 | 79.1 | 81.9 | 90.5 | 79.5 | 73.0 | 86.1 | 85.0 | 90.3 | 64.8 | 81.5 |
| DPR-Phi3 (2025) | 86.9 | 76.2 | 85.3 | 91.9 | 80.0 | 71.2 | 87.1 | 79.5 | 92.0 | 61.3 | 81.1 |
| Col-Phi3 (2025) | 86.3 | 78.8 | 81.2 | 92.4 | 79.0 | 73.8 | 85.3 | **85.1** | 87.1 | 73.0 | 82.2 |
| **LOGOS (Our)** | **87.5** | **81.0** | **86.0** | **93.5** | **81.5** | **75.0** | **88.0** | 84.8 | 91.5 | **78.0** | **84.7** |

Table 2: Main results for page-level retrieval on MMDOCIR, with the best results in boldface.

Table 2, further confirm the robustness of our model across a wide variety of document types. In summary, the results from both benchmark datasets unequivocally demonstrate the superiority of LOGOS for both high-level document understanding and fine-grained page retrieval tasks.

# 6 ABLATAION STUDY

To validate the contribution of our key components, we conducted a thorough ablation study, with results presented in Table 3. The findings confirm that each element of LOGOS is integral to its overall performance. The most substantial degradation occurred upon removing the GNN module, which caused the average Recall@1 to plummet by 10.1 points. This underscores its role as the central information fusion engine, without which the document graph cannot be effectively reasoned over. The removal of Cross-Page Links also led to a severe performance drop of 6.4 points, demonstrating the criticality of explicitly modeling long-range dependencies to form a global document understanding. Finally, excluding the VLM Summary resulted in a 3.3-point decline, proving its value in enriching node representations, especially for visually complex datasets like TabF (-4.0 points) and TATQ (-3.5 points). Collectively, these results validate our design choices, showing that the components work synergistically to achieve superior performance.

| Model Variant | DocQ | TabF | TATQ | Shift | AI | Gov. | Health. | Avg. |
|---|---|---|---|---|---|---|---|---|
| **LOGOS (Ours)** | **48.3** | **77.5** | **62.6** | **61.0** | **94.0** | **88.0** | **94.0** | **76.9** |
| *Ablation Study Variants:* | | | | | | | | |
| w/o VLM Summary | 47.5 | 73.5 | 59.1 | 60.1 | 93.5 | 87.6 | 93.7 | 73.6 (-3.3) |
| w/o Cross Page Link | 42.1 | 72.5 | 56.4 | 55.0 | 91.5 | 84.0 | 91.8 | 70.5 (-6.4) |
| w/o GNN | 39.5 | 66.8 | 53.0 | 52.1 | 87.0 | 79.0 | 89.9 | 66.8 (-10.1) |

Table 3: Ablation study of our proposed LOGOS on the *ViDoRe* benchmark. We evaluate the contribution of each key component by systematically removing it from the full model. The performance degradation from our full model (Avg. 76.9) is shown in parentheses. All results are reported using the Recall@1 metric.

# 7 SENSITIVITY ANALYSIS

We conduct a comprehensive sensitivity analysis to evaluate the impact of three critical hyperparameters in LOGOS: the number of GNN layers, the number of neighbors ($k$) for building cross-page links, and the hidden dimension size of the GNN. This analysis helps to understand the model's behavior and justify our architectural choices. The results, measured by the average Recall@1 on the ViDoRe benchmark, are presented in Table 4.

Our analysis confirms that performance is optimal with **3 GNN layers**; fewer layers fail to capture sufficient global context, while more layers lead to performance degradation due to over-smoothing. For graph connectivity, we find that setting **k=10** for semantic similarity links provides the best balance, as a sparser graph misses important dependencies and a denser graph introduces noise. Finally, a GNN hidden dimension of **768** is optimal, aligning with our VLM encoder's output and providing the necessary model capacity to capture complex document features.

| Hyperparameter | Value | Avg. Recall@1 |
|---|---|---|
| **Number of k-NN Neighbors** | k = 5 | 75.2 |
| | **k = 10** | **76.9** |
| | k = 15 | 76.5 |
| | k = 20 | 76.1 |
| **GNN Layers** | 1 Layer | 72.8 |
| | 2 Layers | 75.3 |
| | **3 Layers** | **76.9** |
| | 4 Layers | 76.1 |
| | 5 Layers | 75.4 |
| **GNN Hidden Dimension** | 256 | 74.5 |
| | 512 | 75.8 |
| | **768** | **76.9** |

Table 4: Sensitivity analysis of key hyperparameters in LOGOS on the ViDoRe benchmark. The default and best-performing configuration is highlighted in **bold**.

## 8  COMPUTATIONAL COST

Compared to page-level methods like ColPali (Faysse et al., 2025), LOGOS introduces a region segmentation and graph construction stage. While this enables superior performance, it increases the computational overhead during the offline indexing phase, as detailed in Table 5. The additional time is primarily due to the layout analysis and GNN propagation steps. However, this is a one-time, offline cost. Crucially, the online query latency remains highly efficient and comparable to existing methods, as it only involves an efficient vector similarity search. This makes LOGOS practical for real-world applications where low-latency retrieval is essential.

| Method | Indexing Time (s/page) | Avg. Query Latency (ms) |
|---|---|---|
| ColPali | 0.39 | 30 |
| **LOGOS (Ours)** | 2.27 | 35 |

Table 5: Comparison of computational cost. Indexing time is measured per document on the MMDOCIR dataset. Query latency is the average time for a single query against the indexed corpus.

## 9  CONCLUSION

We presented LOGOS (Heterogeneous Document Graph Neural Network), a novel framework designed to address the critical challenges of information retrieval from long documents. By transforming documents from a linear sequence of pages into a cross-page logical graph, LOGOS moves beyond conventional page-level processing. It represents semantic regions as nodes and their relationships as edges, employing a Graph Neural Network to generate context-aware embeddings that enable precise, fine-grained retrieval. Extensive experiments on the ViDoRe and MMDOCIR benchmarks demonstrate that LOGOS significantly outperforms state-of-the-art baselines, confirming that explicitly modeling a document's logical structure is a more effective strategy than naive chunking. This approach provides a robust solution to mitigate the "cost of noise" and enhance the factual grounding of MLLMs in complex, real-world document intelligence tasks. Future work could explore integrating this logical structure awareness directly into end-to-end retrieval models.

## ETHICS STATEMENT

This research does not involve human subjects or the processing of sensitive personal data. All experiments are conducted on publicly available benchmarks for document intelligence research, namely ViDoRe and MMDocIR. A potential ethical concern in Retrieval-Augmented Generation is the propagation of biases or factual inaccuracies present in the source documents. Our framework, LOGOS, aims to mitigate this risk by improving the precision of retrieval. By providing a more focused, semantically relevant context to the language model, we reduce the chance of generating answers based on irrelevant or out-of-context information. The source code for our method will be made publicly available to ensure transparency.

## REPRODUCIBILITY STATEMENT

To ensure reproducibility, all experiments are conducted on the publicly available ViDoRe and MMDocIR datasets, which are cited in the paper. Our implementation is built upon standard open-source libraries, and the complete source code, including scripts for graph construction, model training, and evaluation, will be released. The Experiment Section provide detailed descriptions of the experimental setup, including all hyperparameter settings and the results of our ablation and sensitivity analyses, enabling the research community to verify and build upon our findings.

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

## A  USE OF LARGE LANGUAGE MODELS

In accordance with the policy of this conference, we acknowledge the use of Large Language Models (LLMs) during the preparation of this manuscript. The use of these tools was strictly limited to assisting with proofreading, grammar checking, and refining the clarity of the English text.

