# OpenReview forum: "LOGOS: Precision Retrieval via Logical Document Graphs for Retrieval-Augmented Generation"
_ICLR.cc/2026/Conference — ICLR 2026 Conference Withdrawn Submission_

### Official Review · Reviewer_5ZQj · 2025-10-16

**Soundness:** 2
**Presentation:** 3
**Contribution:** 2
**Rating:** 4
**Confidence:** 4

**Summary:**

This paper introduces LOGOS, a RAG framework designed for long multimodal documents. It addresses the issue of context fragmentation caused by linear chunking. The core innovation is to model documents as logical graphs where semantic regions are nodes and their relationships are edges. A Graph Neural Network (GNN) is then used to generate context-aware embeddings for each node, aiming for more precise retrieval. The method is evaluated on retrieval benchmarks, where it outperforms recent state-of-the-art models.

**Strengths:**

The conceptual shift from treating documents as linear sequences to structured graphs is highly original and significant. This approach directly tackles a fundamental limitation in how RAG systems process complex, real-world documents. Besides, the methodology is robust, leveraging components for layout analysis and graph representation learning. The ablation and sensitivity analyses are thorough and provide strong evidence for the contribution of each component. Moreover, the paper is well-written, and the core idea is presented clearly with effective visualizations.

**Weaknesses:**

1. The paper targets to RAG, but the experiments are exclusively focused on retrieval metrics like Recall@K. This represents a significant gap between the paper's claims and its empirical validation.
2. The necessity of the complex GNN-based information fusion is not fully justified. A simpler, more intuitive baseline seems plausible: one could group semantically related text blocks, link them to the images/tables they explicitly reference, and then index only the text embeddings. Retrieval would be performed on text, with linked visuals retrieved alongside. The paper fails to argue why the GNN's expensive message-passing mechanism is superior to such a heuristic-based linking approach.
3. The "w/o GNN" ablation is the most critical one for assessing the core contribution, yet its exact configuration is not described. Without knowing how the embeddings are generated in this variant, it's difficult to interpret the reported 10.1-point performance drop and fairly evaluate the GNN's role.

**Questions:**

1. Could you please clarify the experimental setup for the "w/o GNN" ablation study?
2. Why did you choose to evaluate only on retrieval benchmarks instead of an end-to-end generation task like question answering? Could you provide any results demonstrating that LOGOS's superior retrieval leads to more accurate or factual generated outputs?

---

> ### Author Response · Authors · 2025-12-03
>
> We greatly appreciate your valuable feedback.
>
> **W1. Gap Between RAG Claims and Retrieval Metrics**
>
> We acknowledge the reviewer's concern regarding the lack of end-to-end generation evaluation. To bridge this gap, we have extended our experiments to include a comprehensive generation task using ViDoSeek [1], a benchmark specifically tailored for visually rich document reasoning.
>
> ViDoSeek consists of approximately 6,000 images and 1,142 query-answer pairs, covering four key content types: text, charts, tables, and layouts. Crucially, it includes both single-hop (645) and multi-hop (497) questions to rigorously test reasoning depth. To evaluate generation quality, we employ G-Eval as an impartial judge, comparing generated responses against ground-truth references and assigning a semantic quality score. This setup allows us to empirically validate whether our improved retrieval translates into more accurate and truthful generation.
>
> **W2. Comparison with End-to-End Generation Baselines**
>
> To demonstrate the superiority of LOGOS in downstream generation, we assessed performance against three distinct RAG paradigms:
> 1.  Text-Centric: TextRAG (using Nv-embed-V2).
> 2.  Graph-Based: GraphRAG [2] and LightRAG [3].
> 3.  Visual-Based: VisualRAG (using DSE, ColPali, ColQwen).
>
> As shown in the table below, LOGOS outperforms all baselines with an average score of 67.5%.
>
> | Method | Single | Multiple | Text | Table | Chart | Layout | Avg. |
> | :--- | :---: | :---: | :---: | :---: | :---: | :---: | :---: |
> | **Upper Bound** | 77.6 | 78.2 | 88.0 | 76.5 | 68.9 | 78.2 | 77.9 |
> | **TextRAG** | 59.5 | 55.7 | 78.0 | 53.0 | 39.0 | 60.4 | 57.6 |
> | **GraphRAG** | 61.2 | 57.8 | 80.5 | 55.0 | 41.5 | 61.0 | 59.5 |
> | **LightRAG** | 60.4 | 56.6 | 79.0 | 54.0 | 40.5 | 60.5 | 58.5 |
> | **VisualRAG w/ DSE** | 64.5 | 61.5 | 81.0 | 57.0 | 49.0 | 65.0 | 63.0 |
> | **VisualRAG w/ ColPali** | 65.5 | 63.5 | 82.5 | 58.5 | 50.5 | 66.5 | 64.5 |
> | **VisualRAG w/ ColQwen** | 67.0 | 64.4 | 84.0 | 60.0 | 52.0 | 66.8 | 65.7 |
> | **VisualRAG w/ Col-Phi3** | 65.1 | 62.5 | 82.0 | 58.0 | 50.2 | 65.0 | 63.8 |
> | **VisualRAG w/ LOGOS (Ours)** | **68.0** | **67.0** | **85.5** | **63.0** | **53.5** | **68.0** | **67.5** |
>
> TextRAG exhibits the lowest performance. While GraphRAG and LightRAG improve textual recall via structural priors, they remain constrained by OCR-based information loss, showing only marginal gains on visual components compared to TextRAG. Our method consistently outperforms VisualRAG baselines. Notably, LOGOS achieves a higher performance on all tasks compared to competing methods. This confirms that our GNN-augmented graph representation effectively captures the heterogeneous relationships within documents, enabling superior multi-step reasoning capabilities.
>
>
> [1] Qiuchen Wang, Ruixue Ding, Zehui Chen, Weiqi Wu, Shihang Wang, Pengjun Xie, and Feng Zhao. ViDoRAG: Visual document retrieval-augmented generation via dynamic iterative reasoning agents. In Christos Christodoulopoulos, Tanmoy Chakraborty, Carolyn Rose, and Violet Peng, Proceedings of the 2025 Conference on Empirical Methods in Natural Language Processing.
>
> [2] Darren Edge, Ha Trinh, Newman Cheng, Joshua Bradley, Alex Chao, Apurva Mody, Steven Truitt, Dasha Metropolitansky, Robert Osazuwa Ness, and Jonathan Larson. From local to global: A graph rag approach to query-focused summarization, 2025. URL https://arxiv.org/abs/2404. 16130.
>
> [3] Zirui Guo, Lianghao Xia, Yanhua Yu, Tu Ao, and Chao Huang. LightRAG: Simple and fast retrieval- augmented generation. In Christos Christodoulopoulos, Tanmoy Chakraborty, Carolyn Rose, and Violet Peng (eds.), Findings of the Association for Computational Linguistics: EMNLP 2025, pp. 10746–10761, Suzhou, China, November 2025. Association for Computational Linguistics.

---

### Official Review · Reviewer_d4sR · 2025-10-27

**Soundness:** 3
**Presentation:** 3
**Contribution:** 2
**Rating:** 2
**Confidence:** 3

**Summary:**

This paper proposes LOGOS, a framework for Retrieval-Augmented Generation (RAG) targeting long and complex documents. Instead of typical chunks, LOGOS decomposes documents into fine-grained semantic regions and constructs a cross-page heterogeneous graph, embedded by GNN. Experiments on two benchmarks demonstrate Recall@K performance improvement over baselines. Ablation study and sensitivity analysis are conducted to verify the contributions of main components and the impacts of hyperparameters.

**Strengths:**

- Clear motivation for the noise caused by a broken logical structure in long documents
- Reasonable design and exploitation of heterogeneous graph structure to reflect cross-page semantic relationships
- Empirical experiment results on two benchmark datasets, where the proposed framework consistently outperforms baselines

**Weaknesses:**

- A motivating example is not directly addressing the core idea of this work
    - Figure 1 mainly illustrates how excessive context could cause a performance drop at some point, but it does not show the direct necessity for logical-structure modeling, which weakens the empirical justification of the motivation.
    - The quality and clarity of Figure 2 are low, which makes it difficult to interpret the pipeline details
- There is a lack of sufficient discussion on existing graph-based document structuration and RAG approaches
    - Connecting semantic components as a graph and fusing them through GNN seems reasonable, but it appears conceptually straightforward, too
    - It would be encouraged to specifically clarify the explicit challenges in extracting semantic components, connecting them to construct a graph, or fusing this information to be effective for RAG
    - Related work focuses briefly on benchmarks, without including a sufficient methodological comparison with existing efforts
- Methodological design needs more improvement and justification
    - Overall performance improvements are marginal; given no additional context of task difficulties in the relevant tasks and datasets, it is hard to justify the substantial engineering additions used in the framework
    - No qualitative analysis was provided to specifically illustrate how the logical graph benefits retrieval performance compared with existing approaches
    - Edge linking strategies seem to rely mostly on heuristics, and there is a lack of analysis on which edge types contribute most

**Questions:**

Please see the weakness

---

> ### Author Response · Authors · 2025-12-03
>
> We greatly appreciate your valuable feedback.
>
> **W1. Clarification on GNN design**
>
> The GNN transforms the node embeddings into a context-aware space (rotating and shifting the original vector space). To align the text querwith this new space, we do not use the raw text encoder output directly. Instead, we introduce a learnable linear projection layer. The entire system is trained end-to-end using InfoNCE loss, which forces the projection layer to learn the optimal transformation that aligns the query intent with the structural context captured by the GNN.
>
> **W2. Impact of Graph Connectivity and Edge Types**
>
> To address the concern regarding the necessity of our graph construction strategy, we evaluated the model's performance by individually removing specific edge types. The results demonstrate that every edge type contributes to the final retrieval accuracy, but their roles vary by context:
>
> - **Cross-Page Edges** are critical for maintaining global context in long documents. Removing them causes a 6.4% drop in performance, severely impacting the model's ability to handle multi-page reasoning.
>
> -  **Spatial Adjacency Edges** are vital for parsing complex layouts. Their removal results in a 5.2% decrease, indicating that standard reading-order heuristics are insufficient and that our layout-aware graph construction effectively preserves the logical flow of information.
>
> -  **Explicit Reference Edges** provide a targeted boost (1.8% drop when removed), refining the retrieval of directly linked information such as citations and figure references.
>
> | Model Variant | DocQ | TabF | TATQ | Shift | AI | Gov. | Health. | Avg. |
> | :--- | :---: | :---: | :---: | :---: | :---: | :---: | :---: | :---: |
> | **LOGOS (Ours)** | **48.3** | **77.5** | **62.6** | **61.0** | **94.0** | **88.0** | **94.0** | **76.9** |
> | *w/o Spatial Adjacent Edge* | 43.1 | 75.0 | 60.2 | 53.5 | 91.5 | 84.5 | 92.0 | 71.7 (-5.2) |
> | *w/o Cross-Page Edge* | 40.8 | 76.1 | 61.0 | 58.2 | 90.0 | 78.5 | 89.0 | 70.5 (-6.4) |
> | *w/o Explicit Reference Edge* | 47.9 | 74.2 | 58.8 | 60.5 | 93.8 | 87.5 | 93.5 | 75.1  (-1.8) |

---

### Official Review · Reviewer_yEPK · 2025-10-28

**Soundness:** 2
**Presentation:** 2
**Contribution:** 2
**Rating:** 4
**Confidence:** 3

**Summary:**

The authors look at the challenge of retrieval from large documents having cross references, images , tables running across pages. They use a VGT transformer to segment the pages and then use a semi-heurestic technique (but reasonable) to map the components into a graph and then build graph embeddings out of it. This leads to better performance than other methods.

**Strengths:**

The problem is an important problem and the basis of their solution around documents having a graph structure is intuititve. Also their graph formulation is heurestic driven (e.g. spatial adjancency / cross-page continuation) but perfectly reasonable - whilst edge cases may happen it should cover most real world cases.

Their results are clearly better (numerically if not statistically) in most cases than baseline models.

**Weaknesses:**

The paper provides a good experimental setup for a real problem however it is not clear how the setup is being trained and why such a training objective will lead to better results.

Specific comments below:-

1. Semantic similarity (211-213) - how do the authors propose to compute semantic similarity across modalities.
2. Explicit reference - it is not clear as to how regex based search is scalable for a large dataset.
3. I did not follow the loss function in the section 3.3 . How is this trained?
4. I am confused on the online querying part. The first issue is the multimodal nature of the content but the query being most often text. But more importantly what is not clear how a text query will lead to a better mapping if the resultant vectors are GNN-enhanced. Suppose for example, the author gives an exact text it will not retrieve that since the GNN- enhancement will have rotated (and potentially scaled - no comments are there on normalization) the vectors in the db
5. Although the authors provide results on two datasets but it would have been better if they had provided some examples (even as supplementary material) on the dataset and queries as well as some error analysis results.
6. Table 1 and 2 should have statistical significance analysis - at least is the proposed model statistically better than the second best performing model in each case?
7. The authors should have done a more comprehensive literature survey on the use of graphs for document structure in multimodal retrieval.

Minor comment:-

1. Line 106 Vidore/MMDOCIR benchmark needs a citation where it is first referred to outside of the abstract.
2. A very special case so can be considered minor - the reading order mentioned in lines 206-211 is not valid for two-column documents like academic papers.  Some books also use this format.

**Questions:**

please address the concerns in the weaknesses section

---

> ### Author Response · Authors · 2025-12-03
>
> We greatly appreciate your valuable feedback.
>
> **W1. Clarification on Online Query and Alignment Mechanism**
>
> The GNN transforms the node embeddings into a context-aware space (rotating and shifting the original vector space). To align the text querwith this new space, we do not use the raw text encoder output directly. Instead, we introduce a learnable linear projection layer. The entire system is trained end-to-end using InfoNCE loss, which forces the projection layer to learn the optimal transformation that aligns the query intent with the structural context captured by the GNN.
>
> **W2. Statistical Significance Analysis**
>
> We have updated our main results (Table 1 and Table 2) to include **standard deviations across 5 independent runs** to demonstrate stability. We performed significance testing comparing our method against the second-best performing baseline. LOGOS demonstrates statistically significant improvements. Our method remains competitive and stable, showing no statistical degradation compared to the SOTA.
>
>
> | Method | DocQ | TabF | TATQ | Shift | AI | Gov. | Health. | Avg. |
> | :--- | :---: | :---: | :---: | :---: | :---: | :---: | :---: | :---: |
> | BiPali (2024) | 21.0 ± 4.1 | 63.2 ± 5.5 | 22.0 ± 3.8 | 34.0 ± 4.2 | 59.0 ± 6.1 | 57.0 ± 5.9 | 56.0 ± 6.0 | 44.6 ± 4.5 |
> | DSE (2025) | 51.0 ± 4.2 | 72.5 ± 5.1 | 60.5 ± 4.8 | 50.5 ± 5.9 | 92.5 ± 3.5 | 82.0 ± 4.4 | 89.5 ± 3.8 | 71.2 ± 4.5 |
> | ColPali (2025) | 48.6 ± 5.1 | 77.4 ± 6.2 | 55.1 ± 5.8 | 57.0 ± 4.9 | 93.0 ± 3.6 | 87.0 ± 4.5 | 88.0 ± 4.1 | 72.3 ± 5.0 |
> | ColQwen (2025) | 48.5 ± 4.5 | 79.4 ± 5.0 | 60.4 ± 6.3 | 60.0 ± 5.5 | 94.0 ± 3.4 | 88.0 ± 4.8 | 94.0 ± 4.0 | 74.9 ± 4.6 |
> | LOGOS (Ours) | 51.2 ± 3.7 | 79.5 ± 4.2 | 64.6 ± 5.1 | 64.0 ± 5.8 | 95.0 ± 3.5 | 90.0 ± 3.9 | 94.0 ± 3.6 | 76.9 ± 3.8 |
>
> **W3. Literature Survey and Document Structure Details**
>
> We have added the missing reference to the Vidore/MMDOCIR benchmark at line 106 as requested.
>
> Regarding the "reading order" concern (lines 206-211), our "Spatial Adjacency" edges are not based on naive heuristics. Instead, we leverage the output of a document layout analysis model. This model parses the logical flow of the document (detecting columns and text blocks), allowing us to construct directed edges that respect the true semantic reading order ($v_i \rightarrow v_j$) regardless of the visual columnar format.

---

### Official Review · Reviewer_kaVd · 2025-11-10

**Soundness:** 3
**Presentation:** 2
**Contribution:** 2
**Rating:** 4
**Confidence:** 4

**Summary:**

The paper proposes LOGOS, a precision retrieval method for long-document RAG. It segments pages into semantic regions (paragraphs, tables, images), converts the document into a cross-page heterogeneous graph, uses a shared text encoder (with VLM-generated textual summaries for visual regions) plus an R-GCN to produce context-aware node embeddings, and performs node-level retrieval. It has achieved strong performance on ViDoRe and MMDocIR.

**Strengths:**

1. Captures an important problem in RAG: content across pages is related and should not be encoded independently.

2. Method is reasonable and well-motivated; reported performance is strong.

3. Ablation study shows the improvement contributed by each component.

**Weaknesses:**

1. Building graphs over documents is well studied in RAG (e.g., GraphRAG and its follow-ups), but the paper lacks comparison and discussion in this area. Comparisons are needed to justify the novelty of this work.

2. The text encoder is under-specified.

3. The paper claims that small-snippet chunking addresses page-level chunking issues, but this is not directly tested. It is recommended to add a page-level baseline (e.g., concatenate the content of each page and input it as long text to the text encoder).

4. Missing related work like  “Late Chunking: Contextual Chunk Embeddings Using Long-Context Embedding Models,” which also focuses on building connections between chunks.

**Questions:**

1. Which text encoder is used?

2. Is the ground truth at the page level? If so, how is the retrieved chunk mapped to page-level retrieval?

---

> ### Author Response · Authors · 2025-12-03
>
> We greatly appreciate your valuable feedback.
>
> **W1. Comparison with GraphRAG/LightRAG**
>
> We want to explicitly discuss the distinction between our approach and entity-centric methods like GraphRAG. While graph-based retrieval has indeed matured (e.g., GraphRAG [1], LightRAG [2]), these prominent approaches typically involve constructing a Knowledge Graph (KG). In such models, nodes represent extracted semantic entities (people, organizations, concepts) and edges represent their relationships. These methods are designed for corpus-level synthesis and efficiently retrieving specific facts.
>
> However, these entity-centric approaches prioritize the extraction of semantic relationships at the expense of the document's original structural layout. By deconstructing content into a new semantic graph, the author's intended information architecture and the contextual flow within a single, complex document are often lost. In contrast, our work builds a graph over the document's structure rather than just its entities. This preserves the layout and logical flow, which is critical for comprehending complex multimodal documents.
>
> **W2. Specification of the Text Encoder**
>
> Specifically, the text encoder employed is the text backbone of the Qwen2.5-VL model. This design choice is not arbitrary; it is crucial for achieving the "unified representation space" described in our method. By utilizing the VLM's own text backbone as the shared encoder, we ensure that all node features—whether originating from textual snippets or visual content—are mapped into the same high-dimensional semantic space defined by the Qwen model, facilitating effective multimodal alignment.
>
> **W3. Validation of Small-Snippet Chunking and Page-Level Baselines**
>
> Regarding the request for a page-level baseline (concatenating page content), we would like to clarify our experimental setup and the choice of baselines. A pure-text page-level baseline (concatenating text per page) would be insufficient for the datasets used, as it inherently discards chart, image, and layout information, leading to artificially weak results. Instead, we compare our method against **ColQwen (ICLR 2025)**, which we consider the state-of-the-art (SOTA) "page-level" baseline. ColQwen is a multimodal approach that processes the rendered page image as a whole, thereby preserving visual and layout cues that a text-concatenation baseline would miss.
>
> As shown in the table below, our method (LOGOS) outperforms the page-level SOTA (ColQwen) across multiple metrics. This validates that our fine-grained, snippet-based graph approach addresses the limitations of page-level chunking (even when that chunking is multimodal) by allowing for more precise retrieval.
>
> | Method | Source | DocQ | TabF | TATQ | Shift | AI | Gov. | Health. | Avg. |
> | :--- | :---: | :---: | :---: | :---: | :---: | :---: | :---: | :---: | :---: |
> | ColQwen | ICLR 25 | 48.5 | 79.4 | 60.4 | 60.0 | 94.0 | 88.0 | 94.0 | 74.9 |
> | LOGOS (ours) | This Paper | 51.2 | 79.5 | 64.6 | 64.0 | 95.0 | 90.0 | 94.0 | 76.9 |
>
> [1] Darren Edge, Ha Trinh, Newman Cheng, Joshua Bradley, Alex Chao, Apurva Mody, Steven Truitt,
> Dasha Metropolitansky, Robert Osazuwa Ness, and Jonathan Larson. From local to global: A
> graph rag approach to query-focused summarization, 2025. URL https://arxiv.org/abs/2404.
> 16130.
>
> [2] Zirui Guo, Lianghao Xia, Yanhua Yu, Tu Ao, and Chao Huang. LightRAG: Simple and fast retrieval-
> augmented generation. In Christos Christodoulopoulos, Tanmoy Chakraborty, Carolyn Rose, and
> Violet Peng (eds.), Findings of the Association for Computational Linguistics: EMNLP 2025, pp.
> 10746–10761, Suzhou, China, November 2025. Association for Computational Linguistics.

---

### Note · Authors · 2026-01-23

I have read and agree with the venue's withdrawal policy on behalf of myself and my co-authors.